# Silver Itaconate as Single-Source Precursor of Nanocomposites for the Analysis of Chloride Ions

**DOI:** 10.3390/ma15238376

**Published:** 2022-11-24

**Authors:** Tatiana S. Kolesnikova, Anastasiya O. Zarubina, Marina O. Gorbunova, Vladimir A. Zhinzhilo, Gulzhian I. Dzhardimalieva, Igor E. Uflyand

**Affiliations:** 1Department of Chemistry, Southern Federal University, 344090 Rostov-on-Don, Russia; 2Department of Chemistry, Rostov State Medical University of the Ministry of Healthcare of Russian Federation, 344022 Rostov-on-Don, Russia; 3Federal Research Center of Problems of Chemical Physics and Medicinal Chemistry, Russian Academy of Sciences, 142432 Chernogolovka, Russia; 4Moscow Aviation Institute, National Research University, 125993 Moscow, Russia

**Keywords:** metal-containing monomers, single-source precursors, metal nanoparticles, nanocomposites, dynamic gas extraction, digital colorimetry

## Abstract

At present, conjugated thermolysis of metal-containing monomers is widely used as single-source precursors to obtain new metal- and metal oxide-containing nanocomposites. In this study, a detailed analysis of the main stages of conjugated thermolysis of silver itaconate was carried out. The obtained nanocomposites containing silver nanoparticles are evenly distributed in a stabilizing carbon matrix. The structural characteristics and properties of the resulting nanomaterials were studied using X-ray diffraction (XRD), atomic force microscopy (AFM), scanning electron microscopy (SEM), transmission electron microscopy (TEM), and energy dispersive X-ray spectroscopy (EDS). We have developed a method of test analysis of chlorides using paper modified with the obtained silver-containing nanocomposites. The analysis technique is based on the in situ conversion of chlorides to molecular chlorine, its dynamic release, and colorimetric detection using NP-modified paper test strips. A simple installation device is described that allows this combination to be realized. The proposed approach seems promising for nanoparticle-based determinations of other analytes that can be converted into volatile derivatives.

## 1. Introduction

In recent years, metal-containing monomers (MCMs), which have unsaturated bonds or functional groups in molecules that can enter homo-, copolymerization, or polycondensation reactions, have been widely studied [1,2]. The use of MCM in synthesis processes makes it possible to obtain various polymeric materials in one stage, including soft functional materials [3], self-healing and shape memory materials [4], functional nanomaterials [5], and high-performance polymer-supported catalysts [6]. MCMs have been studied most widely on the examples of unsaturated metal carboxylates, the simplest representatives of which are metal acrylates [7,8,9]. In addition, carboxylate complexes have an interesting structure, which is due to diverse ways of binding metal ions by carboxylate groups [8]. Metal salts based on unsaturated dicarboxylic acids are a large group of MCMs which are actively studied by specialists in the field of coordination and polymer chemistry [10,11,12,13]. This is due to the widespread use of maleates, fumarates, itaconates, and acetylenedicarboxylates as coatings with important properties, effective catalysts, magnetic materials, and drugs.

Silver(I) salts based on carboxylic acids have recently become the object of increased attention of researchers in the field of supramolecular chemistry, crystal engineering, and chemistry of coordination polymers [14,15,16,17,18]. In addition, they have antimicrobial and antifungal properties, which makes them objects of study for biochemists and pharmacologists [19,20,21,22,23]. One of the potential uses of Ag(I) complexes as functional materials is the creation of silver-based antiseptic materials. This is due to a much lower propensity to induce microbial resistance compared with antibiotics [24]. In addition, silver(I) ions have surprisingly low toxicity to humans, unlike ions of other heavy metals [25]. It is for this reason that the molecular design of silver(I) complexes is an important aspect of metal-based preparations in bioinorganic chemistry [26]. It should be noted that salts of unsaturated carboxylic acids containing silver attract considerable attention from chemists [27,28,29,30,31,32].

Recently, active research has been conducted on the thermolysis of metal-containing monomers, where MCMs function as single-source precursors for obtaining nanocomposite materials in which metal nanoparticles (NPs) are stabilized by a carbon or polymer shell [33,34]. Silver NPs favorably stand out among various metal NPs because of the manifestation of a wide range of properties: outstanding plasmonic activity, antibacterial activity, chemical stability, good thermal and electrical conductivity, and catalytic activity [35,36,37,38,39]. The scientific approach to such a manifestation is to obtain heterostructured nanomaterials with new properties using environmentally friendly synthesis methods and nontoxic reagents. An undoubted advantage is that the integration of silver NPs into carbon-containing nanomaterials will not only increase their antibacterial efficiency, biocompatibility, and durability but will also reduce their biotoxicity and release into the environment [40].

Previously, we described the use of silver NPs in the analysis of halides [41,42,43,44,45,46,47,48]. In continuation of these studies, this work is devoted to the thermolysis of silver itaconate, the composition and structure of the resulting NPs, and their use for the analysis of chloride ions. The analysis technique is based on the in situ conversion of chlorides to molecular chlorine, its dynamic release, and colorimetric detection using NP-modified paper test strips. In recent years, nanocomposite materials have been widely used to solve environmental problems. However, in most cases, they are used to treat wastewater from heavy metals [49,50,51]. The present study shows the possibility of using the resulting nanocomposite containing silver nanoparticles to remove chlorine and chloride ions from water, which enter water bodies and have a negative effect on them. It should be noted that in this case, water purification occurs with its simultaneous disinfection, which, in our opinion, is a promising direction for providing humanity with safe drinking water. Since it is known that silver(I) salts based on carboxylic acids have antimicrobial and antifungal properties [19,20,21,22,23], the microbiological disinfection of water will occur in the process of cleaning from chemical pollutants. 

## 2. Materials and Methods

### 2.1. Starting Materials

Silver nitrate (AgNO_3_), itaconic acid (C_5_H_6_O_4_), and sodium hydroxide were purchased from Sigma-Aldrich and used without further purification.

### 2.2. Synthesis of Silver Itaconate

All chemical glassware is treated with a mixture of concentrated hydrochloric and nitric acids before use. After draining the acids, the glassware is washed with bidistilled water. All synthesis and auxiliary procedures are carried out exclusively in bidistilled water. Itaconic acid (1.3 g, 0.01 mol) is dispersed in distilled water (50 mL) at a temperature of 50 °C with constant stirring. In a separate vessel, sodium hydroxide (0.8 g, 0.02 mol) is dissolved in 20 mL of distilled water, and after complete dissolution of sodium hydroxide, the latter is poured into a vessel with itaconic acid and stirred for 30 min at 50 °C until the substances are completely dissolved. The resulting solution of sodium itaconate is left at room temperature until it cools down, and 10 mL of a solution containing 3.6 g of silver nitrate AgNO_3_ (0.02 mol) is poured into the cooled solution with constant stirring in the absence of direct sunlight. The mixture is kept on a magnetic stirrer for 1 h in a dark place. The fine crystalline precipitate is filtered off under vacuum, washed with hot water until there are no nitrate ions in the washing water, then washed three times with absolute ethanol in portions of 20 mL, dried initially in the air at 40 °C, and then in dynamic vacuum at 60 °C for 4 h in the absence of light. The amount of 3.26 g of a white fine crystalline powder is obtained, which is 94.7% yield, counting on itaconic acid. Elemental analysis: found C 16.53%, H 1.71%, and Ag 59.82%. Calculated for C_5_H_4_O_4_Ag_2_·H_2_O, C was 16.57%, H 1.65%, and Ag 59.67%.

### 2.3. Characterization Techniques

A CHNOS vario EL cubic analyzer (Elementar Analysensysteme GmbH, Langenselbard, Germany) was used for elemental analysis. Analysis for silver was carried out on an MGA-915 atomic absorption spectrometer (Lumex, St. Petersburg, Russia) or an X-Art M energy-dispersive X-ray fluorescence spectrometer (Comita, St. Petersburg, Russia). The Fourier transform IR (FTIR) spectra were taken with a Perkin Elmer Spectrum 100 FTIR spectrometer (Perkin Elmer, Waltham, MA, USA) from KBr pellets, using Spectrum^TM^ 10 software for the data analysis (Perkin Elmer, Shelton, CT, USA). An STA 409CLuxx synchronous thermal analyzer coupled with a QMS 403CAeolos quadrupole mass spectrometer (NETZSCH, Selb, Germany) and a Perkin-Elmer Diamond TG/DTA derivatograph were used to perform thermal analysis (TA) and differential scanning calorimetry (DSC) in a helium flow (powders, m = 0.3–0.4 g) with standard α-Al_2_O_3_ at a rate of 2 deg/min in the range of 20–500 °C. X-ray diffraction (XRD) analysis was carried out on the diffractometers DRON-UM-2, «Philips PW 1050» and ARL™ X’TRA Powder (Thermo Fisher Scientific, Waltham, MA, USA) with CuKα radiation (λCu = 1.54184 Å) in the range of 2θ = 5–80° angles 2θ with a scan rate of 5°/min and a temperature of 25 °C. The Debye–Scherrer Equation (1) was used to calculate the size of nanomaterial crystallites (D, nm) [52]:(1)$$D=\frac{K\;\lambda \;}{\beta \cos\theta }$$
where K is a constant (ca. 0.9); *λ* is the X-ray wavelength used in XRD (1.5418 Å); *θ* is the Bragg angle; *β* is the pure diffraction broadening of the peak at half-height that is broadening due to the crystallite size.

A ZEISS Crossbeam 340 device (Carl Zeiss) was used to acquire scanning electron microscopic images (SEM) at an accelerating voltage of 3 kV. An Oxford X-max 80 microanalyzer was used to determine the distribution of chemical elements on the surface of samples by X-ray energy-dispersive microanalysis (EDS) with an electron probe energy of ≤10 keV. A high-resolution transmission microscope Tecnai G2 Spirit BioTWIN FEI (Netherlands) was used for high-resolution transmission electron microscopy (HREM). The sample preparation included the following stages: preparation of a powder suspension in hexane, its application to a carbon-coated copper grid, and drying the solvent in air.

The PHYWE Compact AFM was used to perform atomic force microscopy (AFM) in semi-contact mode using a wide-range piezoelectric scanner with the following parameters: lateral scanning up to 100 mm (x–y), scanning in the vertical (z) direction up to 5 mm, scanning speed 1 mm × 1 mm/min, and probe made of single-crystal silicon with aluminum coating with a resonant frequency of 190 ± 60 kHz and a constant hardness of 48 N/m. AFM image analysis was carried out using the Gwyddion 2.10 program [53].

### 2.4. Study of Thermolysis Kinetics

A membrane zero pressure gauge was used to study the kinetics of thermal transformations of MCMs in terms of gas release under static isothermal conditions in an argon atmosphere. The sample weight loss (Δm, wt.%) and the number of gaseous products at ∼20 °C were determined at the end of the experiments.

### 2.5. Preparation of the Nanocomposites

A typical experimental procedure for obtaining nanocomposites is as follows: a weighed portion of silver itaconate (0.6–0.8 g) is placed in a quartz test tube (height of 10 cm and diameter of 2.8 cm), which in turn is placed in a quartz tube sealed at one end 30 cm long and 6 cm in diameter. The assembled device is evacuated to a residual pressure of 6 mm Hg and filled with nitrogen (99.99%) through a hydraulic seal, which uses silicone oil as a barrier fluid. Then, the assembled device is heated to a temperature of 400 °C at a speed of 5°/min. Nitrogen and thermolysis products are pumped out, creating a residual vacuum of 4–6 mm Hg, kept in dynamic vacuum at the specified temperature for 1 h, turned off the heat, and left to reach room temperature in dynamic vacuum. The products are then removed in the form of a porous column 20–25 mm high and crushed. The result is 0.6 g of black powder.

### 2.6. Procedure of Analysis of Chloride Ions

Working solutions of samples were prepared by dissolving their weighed portions in distilled water. Distilled water was obtained using a Millipore Simplicity water purification system (Merk Millipore). A stock solution of chlorides (0.01 mg/mL) was prepared by dissolving 0.0824 g of NaCl in water in a 500.0 mL volumetric flask. Working standard chloride solutions were prepared by diluting appropriate aliquots of the stock solution. The working solutions were placed in a setup for dynamic gas extraction, shown in Figure 1. It consisted of a glass container for the analyzed solution (1), closed with a rubber stopper (2), a test strip holder (3), inside which there was a test strip (4), an air microcompressor (5), connected by a polymer hose (6) with glass bubbler (7), which was inside the vessel. Then, solutions of permanganate and sulfuric acid were added. A Hailea Aco-6601 microcompressor was used to pump air through the reactive system under laboratory conditions. The strips were removed and scanned using Canon CanoScan LiDE 210 (Canon) on a white background with a resolution of 300 ppi. The images obtained in this way were processed in the Adobe Photoshop 7.0 graphics editor in RGB mode by averaging the corresponding color coordinates of individual pixels inside the round, reactive zone. The weight of the reagents was determined on an analytical balance of the 2nd class VLR-20 (Gosmeter) with an error of ±0.0001 g.

### 2.7. Preparation of Paper Test Strips Modified with Ag NPs

The indicator paper was prepared as follows: A solution of Ag NPs (1.5 mL) was placed in a Petri dish. A Whatman Grade 113 paper disc (d = 7.0 cm, m = 0.47 g) was placed in a cup, covering a portion of the solution. In this case, a fairly uniform distribution of the solution over the paper occurred. The paper was then air-dried for 36 h. The operation was repeated 3 times. As a result, paper with a known amount of precipitated Ag NPs (0.62 mg/g) was obtained. The modified paper was cut into test strips. The content of Ag NPs on paper was calculated based on the added total amount of the Ag NP solution.

### 2.8. Calibration

A series of standard solutions was prepared. To do this, the corresponding portion of the initial standard solution of chloride ions (0, 1.5, 3.0, 6.0, 12.0, 18.0, and 24.0 mL) with a concentration of 0.01 mg/mL was placed in a volumetric flask with a capacity of 100 mL and diluted to the mark with distilled water. The final concentrations of chloride ions in the series were: 0, 0.15, 0.3, 0.6, 1.2, 1.8, and 2.4 mg L^−1^, respectively. These solutions were sequentially placed into the reaction vessel of the dynamic gas extraction unit. Concentrated sulfuric acid (2 mL) and 0.2 M KMnO_4_ solution (10 mL) were added to the reaction mixture. The vessel was tightly closed with a rubber stopper with a test strip holder containing a test strip modified with Ag NPs. Next, an air microcompressor was turned on, and the air was bubbled through the solution at a space velocity of 2.8–3.0 L/min for 20 min. After that, the test strip was removed and scanned. The image thus obtained was analyzed in terms of RGB color coordinates.

### 2.9. Analysis of Samples of Natural Water

The analysis of each studied sample was carried out without the stage of sample preparation. The exception was the samples, which had to be diluted to a concentration close to 0.5–1.0 mg/L. A diluted 100 mL solution was placed in a glass vessel of a dynamic gas extraction unit and determined as described in the section above.

### 2.10. Use of Silver Itaconate Thermolysis Product for Adsorption of Chloride Ions

Silica Gel Grade 923 (Sigma Aldrich) was chosen as the carrier. The sorbent is preliminarily sieved, taking a fraction of 0.25–0.5 mm, washed with bidistilled water until there is no reaction to chlorides and sulfates, dried at 120–140 °C, and cooled to room temperature. The sorbent prepared this way is mixed with the thermolysis product of silver itaconate preliminarily dispersed in water using ultrasound (80 W, 15–20 min). The content of the product in relation to silica gel is 3%. The resulting wet sorbent is dried for 4 h at 140 °C and activated at 170 °C in a vacuum for 6 h. Cooled in a vacuum and filling the adsorption column, the height of the combined sorbent layer is 15 cm. Pure prepared silica gel and activated carbon without additional preparation were used as background sorbents. Pure tap water treated with chlorine was passed through the layer of sorbents. Previously, the water sample was analyzed to determine the content of chlorides and residual chlorine. After passing tap water through the corresponding column, the content of chloride ions and residual chlorine in the water was again determined.

## 3. Results and Discussion

### 3.1. Synthesis and Characterization of Silver Itaconate

In the present work, silver itaconate was obtained by direct reaction of silver nitrate with itaconic acid in water in an alkaline medium. FT-IR, TGA, DSC, and XRD were first employed to confirm the structure and physical phase of silver itaconate. Figure 2 shows the IR spectrum of silver itaconate. An intense but rather narrow peak in the region of 3418 cm^−1^ corresponds to the vibrations of the hydroxo group in the composition of the water of crystallization. The sharp peak at 1561 cm^−1^ is attributed to the asymmetric stretch vibration of C=O, whilst the peak at 1385 cm^−1^ is from the symmetric stretch vibration of C=O. Δν is equal to 176 cm^−1^, which may indicate the bidentate mode of coordination of the metal–carboxyl group bond (C_2ν_ symmetry). A weak absorption signal in the region of 644 cm^−1^ corresponds to the metal–oxygen bond.

Figure 3 gives the XRD pattern of silver itaconate. No diffraction peaks from any other impurities were detected, indicating that the reaction was completed, and the product was prepared successfully. In addition, the XRD pattern is consistent with previously published data [27].

Figure 4 shows the TG curve for the decomposition of silver itaconate. Two characteristic regions can be distinguished in the thermal behavior of silver itaconate. The Section 1 lies in the temperature range from 210 to 225 °C and is characterized by a relatively smooth change. On the TGA curve, it corresponds to the weight loss of the substance, which is 5.2%, and may correspond to the loss of crystallization water (theoretical calculation is 4.9%). This is the dehydration stage. The Section 2 begins immediately after the end of the Section 1 and is visualized as an area of intense gas release; on the TGA curve, it looks like an area of abrupt weight loss of a substance in the temperature range from 225 to 235 °C. The total weight loss in the two stages is 30.2%, and considering the first stage, the weight loss in the second stage is 25%, which may correspond to the decarboxylation stage (theoretical calculation 25.58%), which is visualized as intense gas evolution.

On the DSC curve (Figure 5), in the temperature range of 225–234 °C, a single peak is fixed, corresponding to the exothermic effect of 9.4 W/g, which may correspond to the total thermal effect of two rapidly occurring processes—decarboxylation and the polymerization stage following it, followed by deep destructive changes.

In the same temperature range, there is an intensive release of gaseous products (Figure 6).

The dependence of the logarithm of the thermolysis rate constant on inverse temperature for silver itaconate is shown in Figure 7. The activation energy of the process E_a_ = 589 kJ/mol was calculated by the graphical method.

### 3.2. Study of Silver Itaconate Thermolysis Products

The product, which was obtained from the thermolysis of silver itaconate, is a shiny black powder. The color is easily explained by the presence of amorphous carbon [54]. The powder was studied by XRD, which made it possible to obtain data on its phase composition and crystallite sizes (Figure 8). Several Bragg reflections in the picture correspond to Ag with a face-centered cubic (fcc) structure, which is consistent with reports in the literature (JCPDS 4-783). It should be noted that the slight broadening of the XRD peaks is associated mainly with the small particle size. In addition, there are no impure peaks in the XRD pattern because there are no side products other than gas for the AgNO_3_ thermal decomposition reaction.

The elemental composition according to X-ray fluorescence analysis without destroying the object (Figure 9) was C 11.14% and Ag 88.86%, and according to chemical analysis, C 14.06%, H 1.56%, and Ag 84.37%.

Analysis of the SEM image of silver itaconate thermolysis products makes it possible to establish a morphological feature associated with the fact that spherical formations with sizes from 6.8 to 72 nm are visualized on the surface against the background of hollow carbon nanotubes with a diameter of 10 to 42 nm (Figure 10).

The TEM results (Figure 11a) show the presence of predominantly spherical silver particles ranging in size from 6.4 to 72.2 nm (Figure 11b). However, it should be noted that large silver particles are represented by aggregates.

The AFM image (Figure 12) visualizes elongated tubular objects up to 2 µm long and up to 10 nm in diameter, inside which high-density spherical particles are visible on the topographic image. Large particles are segmented over the surface, which suggests the aggregation of small NPs into larger ones under the influence of high temperatures during thermolysis. An analysis of the surface roughness shows that the material is characterized by a clearly defined relief; individual local microheterogeneities reach a height of 50 nm and have an elongated shape.

### 3.3. Determination of Chloride Ions

The preparation of paper test strips with Ag NPs is based on processing paper with portions of a colloidal silver solution. The advantages of this method, of course, are simplicity, the possibility of fairly accurate control of the amount of Ag NPs on paper, the speed of the procedure, which is a consequence of the rapid distribution of the Ag NP solution over the paper, and the rapid evaporation of water at 80 °C.

The color change of the reaction zone was monitored by scanning the Ag NP-modified paper after analyzing and calculating the corresponding RGB color coordinates using conventional imaging software. It has been established that an increase in the concentration of chlorine ions in the analyzed solution leads to an increase in all three-color coordinates. It has also been proven that the dependence of the RGB color coordinates on the concentration of chlorine ions can be adequately described by a first-order exponential equation [52]:y = y_0_ + A(1 − e^−c/t^)
where y is the R, G, or B color coordinate, c is the concentration of an analyte, and y_0_, A, and t are regression parameters.

The calculated regression parameters for calibration curves are given in Table 1. According to [55], a criterion for choosing the most sensitive color coordinate for analytical applications is the maximum A/t ratio.

The A/t ratio is related to the slope of the exponential curves in their initial sections, and, therefore, it determines the limit of detection. Figure 13 shows these ratios for three color coordinates in the RGB system in the case of chloride ions. Obviously, the most sensitive color coordinate is R, and the least sensitive color coordinate is B.

From the given data, it can be seen that for the red coordinate, the maximum value of the “amplitude” of the change in the analytical signal (A1) and the value of A/t, which characterizes the steepness of the graphical dependence, is observed. The A/t value is a criterion for choosing the optimal color channel since the larger its value, the lower the detection area. Thus, according to the proposed method using this paper test strips, the red color coordinate should be chosen as an analytical signal for the colorimetric determination of chlorides (chlorine).

The correctness of the developed method was assessed by the results of the determination of chloride additions in samples of natural waters. To assess the correctness of the measurements, we used the value Θ, which is the difference between the found average value *x* and the sum of the additive and the concentration of chlorides in the analyzed water C (Table 2). The results presented in Table 2 testify to the correctness and repeatability of the proposed method for the determination of chlorides.

### 3.4. Use of the Obtained Nanocomposites for Adsorption of Chloride Ions

The results of the experiment are presented in Table 3. It can be seen from the results that all sorbents are able to reduce the content of chloride ions in water. However, the sorbent obtained by applying the thermolysis product of silver itaconate on silica gel is able to reduce the content of chloride ions to a much greater extent. A somewhat different situation is observed in the case of residual chlorine adsorption. Activated carbon, as a well-known sorbent, most effectively absorbs residual chlorine from water, while silica gel is practically indifferent to chlorine. The sorbent obtained by applying the thermolysis product of silver itaconate on silica gel sorbs chlorine somewhat worse under these conditions. However, its weight is significantly lower than the weight of carbon, which characterizes it as a good sorbent both with respect to chlorine and chloride ions. Silver is known to have antibacterial properties [56,57]. When the purified water passes through a layer of sorbent (silica gel) on the surface of which the thermolysis product of the silver salt is applied, two processes of the interaction of silver with chlorine anions occur simultaneously, and due to the close contact of the purified water (containing microflora) with silver particles, the latter die. In addition, it should be noted that the silver chloride formed during the reaction, which is proved by the appropriate analysis, has more pronounced bactericidal properties. Thus, when the purified water comes into contact with the thermolysis product of the silver salt, a certain amount of chlorine anions contained in the original natural water is bound and disinfected due to contact with nanosized silver particles and the resulting silver chloride. The bactericidal properties of silver chloride are explained by its easy photodissociation with the formation of metallic silver, which exhibits bactericidal properties that are further enhanced by the chlorine formed during the reaction, which also has strong bactericidal properties.

Figure 14 shows the SEM image of the thermolysis product of silver itaconate, which was in contact with a solution of sodium chloride (concentration 0.9%). The morphology of the product has undergone significant changes compared with the initial thermolysis product (see Figure 10).

A relatively loose layer of silver chloride was formed on the surface of silver particles and was also confirmed by the results of the microelement analysis (Figure 15). The weight ratios of silver and chlorine correspond to the molecular formula AgCl (Table 4).

The reason for the interaction of the chlorine anion with silver can be explained by the reaction taking place in the aquatic environment:Ag_n_(H_2_O)_x_ + [Cl^−^(H_2_O)_z_]^−^ → [Ag_n_Cl(H_2_O)_y_]^−^ + (H_2_O)_x+z−y_(2)

Here, x, y, and z are the number of water molecules associated, respectively, with one adsorption center on the metal surface, the chlorine anion located at such a center, and the first hydration sphere of the anion in the bulk of the solution.

A comparison of the proposed method with other methods described in the literature (Table 5) shows that it has good analytical characteristics. This proves good prospects of the proposed approach based on dynamic gas extraction towards solving a task of chloride ions determination in various samples.

## 4. Conclusions

Analysis of the data presented shows that conjugated thermolysis of silver itaconate in an inert atmosphere leads to the formation of metal–carbon nanocomposites containing metal nanoparticles uniformly distributed in a stabilizing carbon matrix. The resulting nanomaterials are stable; during their long-term storage, no changes in the chemical composition, size, and shape of nanoparticles are observed. The proposed method for obtaining nanoparticles is simple and inexpensive, which makes it suitable for large-scale production. It is established that the size of silver crystallites varies from 6.8 to 72 nm on the surface against the background of hollow carbon nanotubes with a diameter of 10 to 42 nm. The combination of Ag NP-based colorimetric detection with dynamic gas extraction has been shown to be able to detect chloride ions in a variety of samples without sample pretreatment. The developed method for the analysis of chloride ions is efficient, simple to perform, and fairly cheap.

## Figures and Tables

**Figure 1 materials-15-08376-f001:**
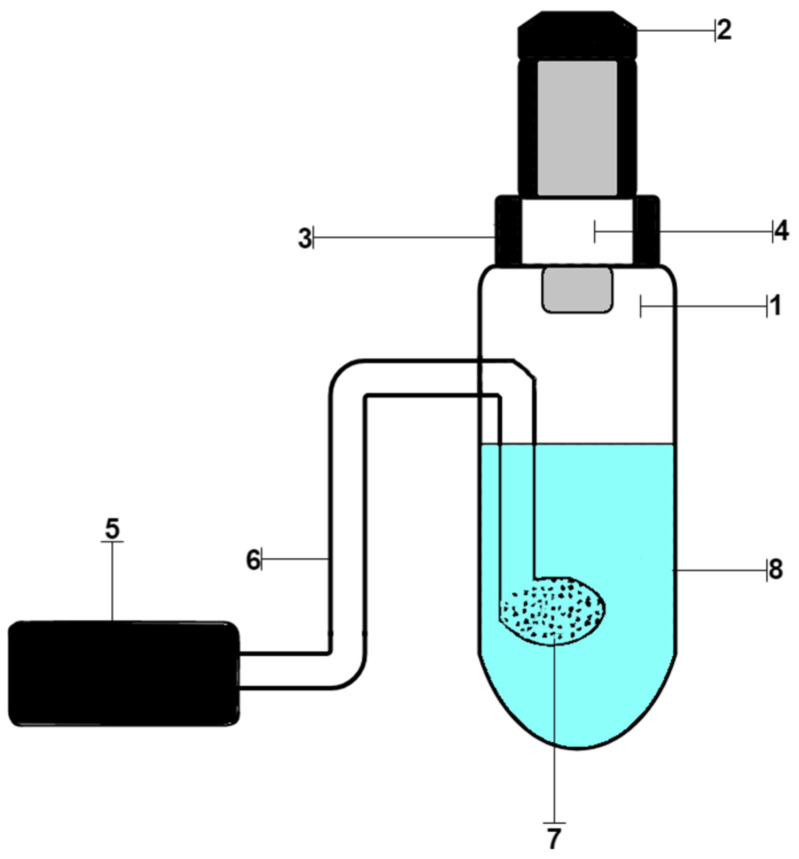
Device for processing dynamic gas extraction: (1) glass container for the analyzed solution; (2) rubber stopper; (3) test strips holder; (4) test strip; (5) air microcompressor, (6) polymer hose; (7) glass bubbler; (8) reaction mixture.

**Figure 2 materials-15-08376-f002:**
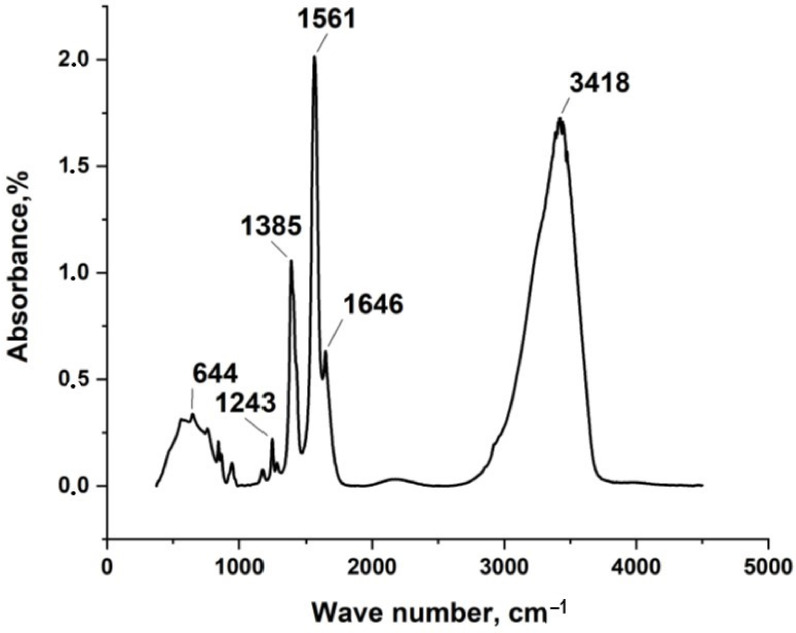
IR spectrum of silver itaconate.

**Figure 3 materials-15-08376-f003:**
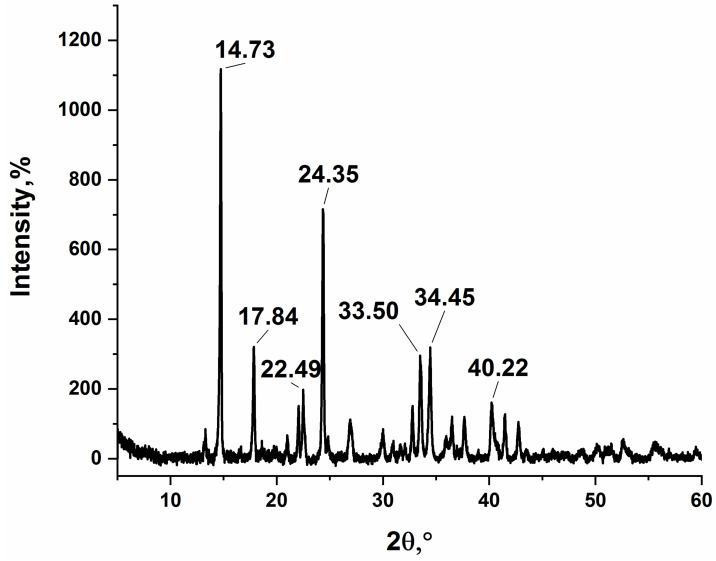
XRD pattern of the obtained silver itaconate.

**Figure 4 materials-15-08376-f004:**
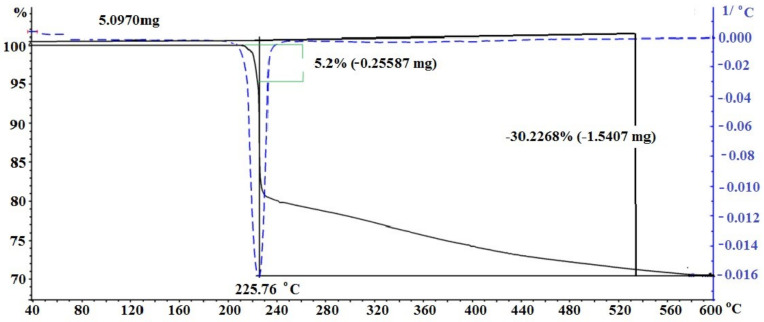
TGA of silver itaconate.

**Figure 5 materials-15-08376-f005:**
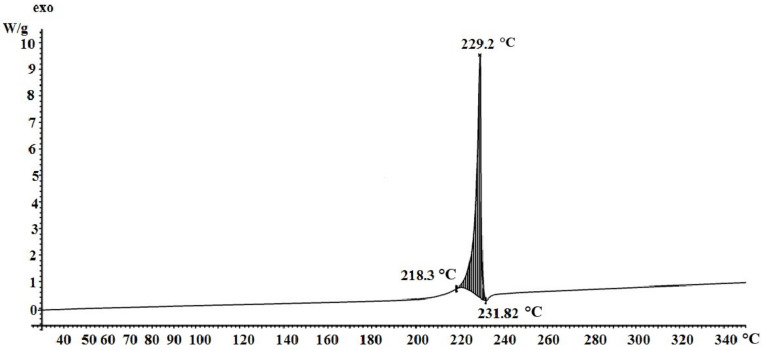
DSC of silver itaconate.

**Figure 6 materials-15-08376-f006:**
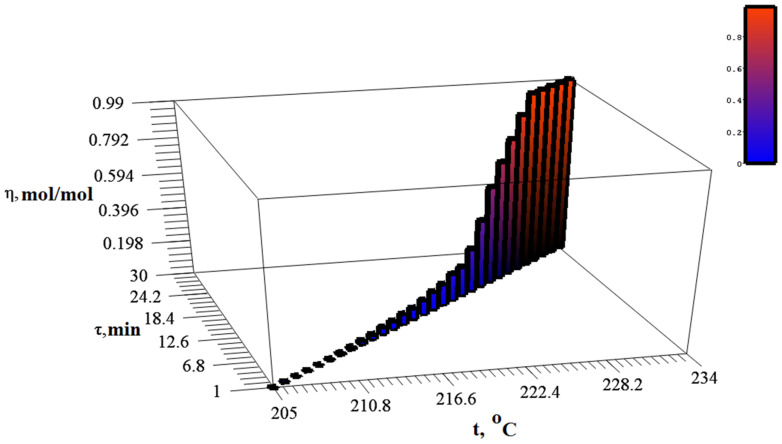
Dynamics of changes in the degree of silver itaconate conversion, depending on time and temperature.

**Figure 7 materials-15-08376-f007:**
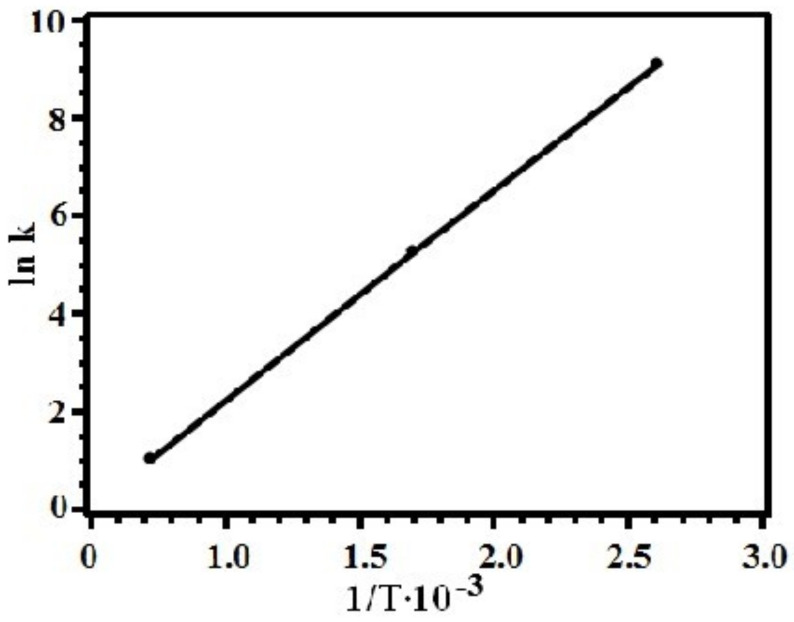
The logarithm of the silver itaconate decarboxylation rate constant versus reciprocal temperature.

**Figure 8 materials-15-08376-f008:**
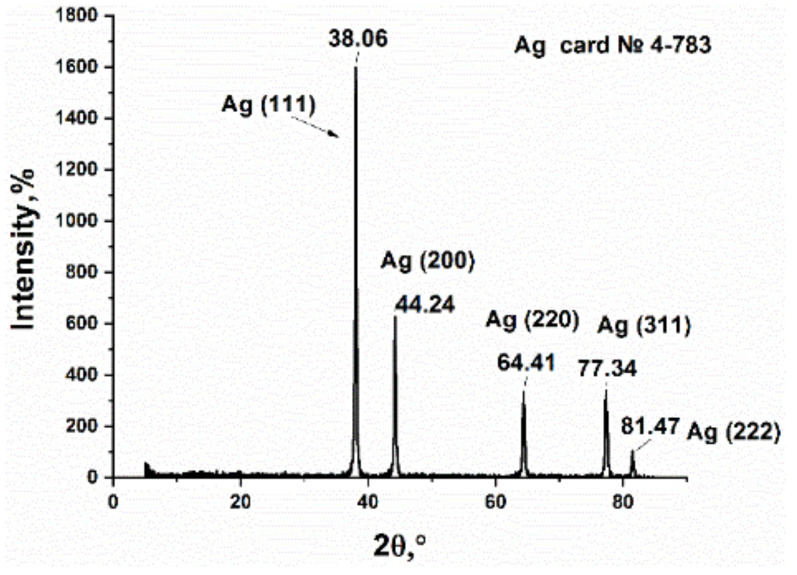
XRD pattern of silver itaconate thermolysis product.

**Figure 9 materials-15-08376-f009:**
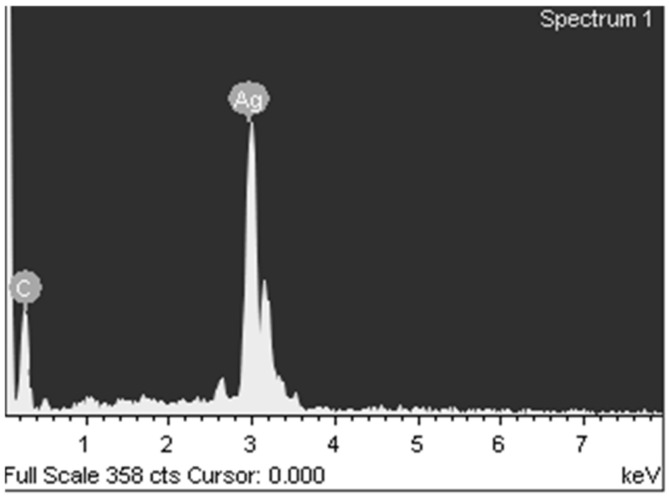
X-ray fluorescence analysis of silver itaconate thermolysis product.

**Figure 10 materials-15-08376-f010:**
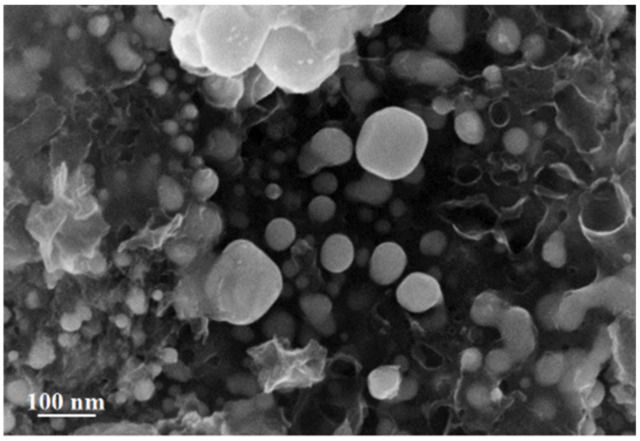
SEM image of silver itaconate thermolysis product.

**Figure 11 materials-15-08376-f011:**
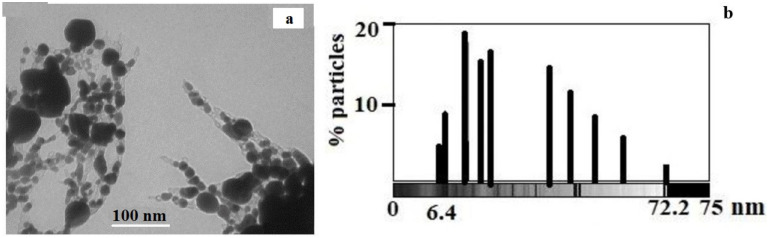
(**a**) TEM image of silver itaconate thermolysis product; (**b**) histogram illustrating the size distribution of silver NPs in the thermolysis product of silver itaconate (**b**).

**Figure 12 materials-15-08376-f012:**
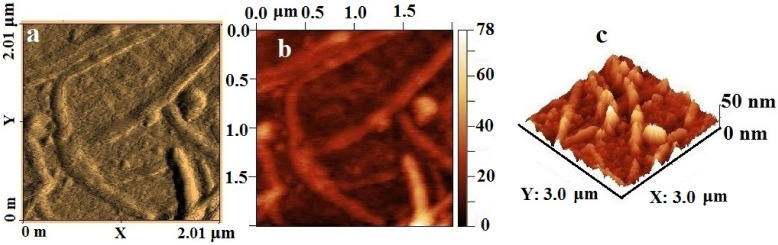
AFM image of silver itaconate thermolysis product: (**a**) amplitude (scan forward), (**b**) topography (scan forward), and (**c**) surface roughness.

**Figure 13 materials-15-08376-f013:**
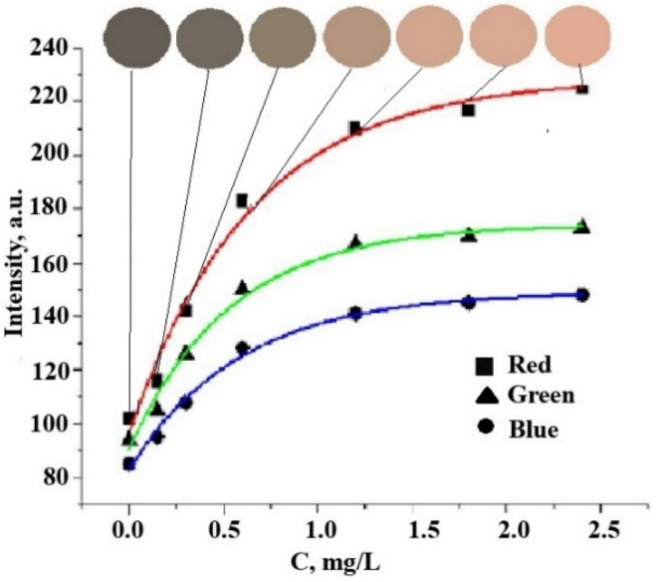
Changes in color and R-, G-, and B-color coordinates of Ag NPs-modified paper on the concentration of chloride ions in the analyzed solution.

**Figure 14 materials-15-08376-f014:**
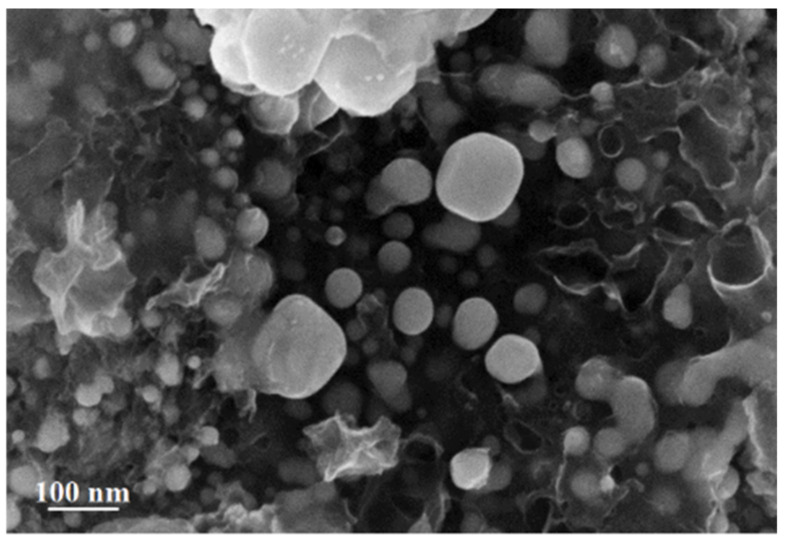
SEM image of the thermolysis product of silver itaconate after treatment with sodium chloride solution.

**Figure 15 materials-15-08376-f015:**
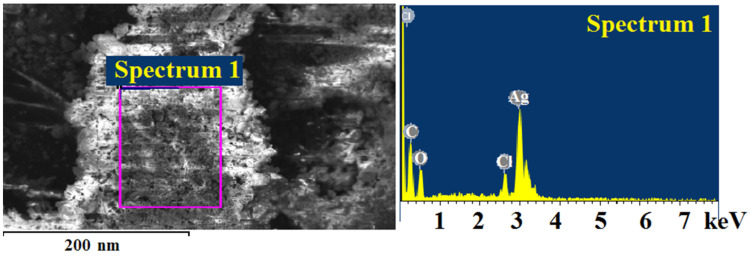
Elemental analysis data on the surface of the thermolysis product of silver itaconate after contact with sodium chloride solution.

**Table 1 materials-15-08376-t001:** Regression parameters and squared correlation coefficients for calibration curves used for the determination of chloride ions.

	Red Coordinate (R)	Green Coordinate (G)	Blue Coordinate (B)
y_0_	96 ± 3	91 ± 3	83 ± 2
A	132 ± 3	84 ± 3	66 ± 3
t	0.65 ± 0.12	0.54 ± 0.09	0.59 ± 0.08
A/t	203	156	111
R	0.9893	0.9899	0.9939

**Table 2 materials-15-08376-t002:** Validation of the chlorides determination results by the “introduced-found” method.

Found (Don River), mg/L	Introduced, mg/L	Found, mg/L	Θ = *x* − C
134.8	10.0	144.0	−0.8
128.0	10.0	139.3	1.3
131.3	10.0	144.5	3.2
130.6	10.0	138.7	−1.9

**Table 3 materials-15-08376-t003:** Results of experiments on the adsorption of chloride ions on various adsorbents.

	Tap Water	Adsorbent		
		Silica Gel	Activated Carbon	Thermolysis Product of Silver Itaconate on Silica Gel
Column flow rate, mL/min		13.2	16.8	14.1
Chlorides, mg/L (decrease, %)	325.83 ± 0.10	296.23 ± 0.05 (−8.83%)	312.23 ± 0.18 (−4.23%)	122.48 ± 0.05 (−62.05%)
Residual chlorine, mg/L (decrease, %)	1.40 ± 0.05	1.30 ± 0.05 (−1.1)	0.03 ± 0.01 (−73.4)	0.04 ± 0.01 (−68.46)
Electrical conductivity, µS/m^2^	1085	1025	1150	775
pH	6.8	6.5	7.2	7.3

**Table 4 materials-15-08376-t004:** The content of elements (wt.%) on the surface of the thermolysis product of silver itaconate after contact with a solution of sodium chloride.

Element	App	Intensity	Weight%	Weight%	
	Conc.	Corrn.		Sigma	
C K	15.46	1.4685	13.28	1.24	38.03
O K	10.41	0.7641	17.19	1.64	36.94
Cl K	3.65	1.0459	4.41	0.60	4.27
Ag L	44.41	0.8605	65.12	1.89	20.76
Totals			100.00		

**Table 5 materials-15-08376-t005:** Comparison of the proposed method with other methods for the determination of chlorine.

Method	LOD, μM	Determination Range, μM	Reference
Voltammetry (screen-printed electrodes)	10,000	10,000–100,000	[58]
Potentiometry (flow-based)	–	1000–6000	[59]
Voltammetry (screen-printed electrodes)	200	200–700	[60]
Spectrophotometry (flow-based)	20	56–560	[61]
Ion chromatography	8.5 μmol kg^−1^	27–850 μmol kg^−1^	[62]
Fluorimetry	0.003	0.01–50	[63]
Present method	1.5	4.2–67.6	This article

## Data Availability

Not applicable.

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
