# Peer review of "Silver Itaconate as Single-Source Precursor of Nanocomposites for the Analysis of Chloride Ions"

_materials, 2022, doi:10.3390/ma15238376_

Round 1

Reviewer 1 Report

In this study, detailed analysis of the main stages of conjugated thermolysis of silver itaconate was carried out by the authors. Herewith the authors have developed a method of test analysis of chlorides using paper modified with the obtained silver-containing nanocomposites. However, numerous concerns are still presented in this manuscript, so I think this manuscript is not suitable for publication in the Materials. The manuscript may be reconsidered after the follow issues are solved.

1. The paper contains 15 groups of figures and 4 tables, please rearrange the charts including but not limited to the format, and modify and explain according to the corresponding format and requirements.

2. As an application-oriented test paper, the conclusion mentions that its structure is stable under long-term storage, what are the specific storage conditions?

3. Compared with conventional chloride detection, how to define the advantages of this method? According to the preparation and synthesis detection method described in the paper, it seems that it does not meet the developed method for the analysis of chloride ions which is efficient, simple to perform, and fairly cheap.

4. Is this method specific to detect chloride ions? Please give a description of the competitive experiment.

Reviewer 2 Report

Review of the Materials-2009812for the Authors:

This article deals with the single-source precursor synthesis for nanocomposites. The article is interesting and does offer enough novelty, but some things need to be revised/checked before it merits publication, so for now my recommendation is minor revision, please see the details below:

Title – Ok

Abstract – Ok.

Introduction – Mostly ok.

“It should be noted that in this case, water purification occurs with its simultaneous disinfection, which, in our opinion, is a promising direction for providing humanity with safe drinking water.” How exactly do you do simultaneous disinfection and purification with your printed samples?

Materials and Methods – Detailed and sufficient.

Results and discussion

XRD-please add a reference or ICDD PDF card number for silver itaconate on the diffractogram.

Also, please check the formatting of the pictures in the whole manuscript, because it seems to be mixed up a bit. Also figure 9 and 11 b could be upgraded, to a level deserving of a scientific publication.

For the AFM pictures, are you sure these were done in Gwyddion and not in the AFM software? The blue colour representation for the topography is also not very common, I would suggest the gold/brown standard. And maybe add the phase scan, instead of the amplitude scan, and also maybe the value of surface roughness?

For the adsorption experiments why is the flow for Ag (4) so much lower than for the other two (13 and 16)?

Conclusions – Ok.

Literature – Ok, maybe just check the proper formatting.

Round 2

Reviewer 1 Report

The author modified the article, added the bacteriostatic principle of this method, gave the AFM image of the material, and listed some conditions of the detection method at the end of the article to illustrate the advantages of this method. Agree to accept the author's article.